# Assessing the Feasibility and Acceptability of a Primary Care Socio-Ecological Approach to Improve Physical Activity Adherence among People with Type 2 Diabetes: The SENWI Project

**DOI:** 10.3390/healthcare11131815

**Published:** 2023-06-21

**Authors:** Guillem Jabardo-Camprubí, Anna Puig-Ribera, Rafel Donat-Roca, Pau Farrés-Godayol, Sebastian Nazar-Gonzalez, Mercè Sitjà-Rabert, Albert Espelt, Judit Bort-Roig

**Affiliations:** 1Faculty of Health Science at Manresa, University of Vic-Central University of Catalonia, Av. Universitaria 4-6, 08242 Manresa, Spain; 2Sports and Physical Activity Research Group, Institute for Research and Innovation in Life and Health Sciences in Central Catalonia (Iris-CC), Ctra. De Roda Núm. 70, 08500 Vic, Spain; 3Sport Exercise and Human Movement (SEaMH), Faculty of Health Science at Manresa, University of Vic-Central University of Catalonia, Av. Universitaria 4-6, 08242 Manresa, Spain; 4Research Group on Methodology, Methods, Models and Outcome of Health and Social Sciences (M3O), Faculty of Health Sciences and Welfare, University of Vic-Central University of Catalonia, Sagrada Familia 7, 08500 Vic, Spain; 5Department of Physical Therapy, Faculty of Health Science Blanquerna, Ramon Llul University, Padilla, 326-332, 08022 Barcelona, Spain; 6Faculty of Health Science Blanquerna, Global Research on Wellbeing (GRoW) Research Group, Ramon Llull University, Padilla, 326-332, 08022 Barcelona, Spain; 7Departament de Psicologia i Metodologia de les Ciències de la Salut, Universitat Autònoma de Barcelona, CIBER de Epidemiologia i Salud Pública, 08193 Barcelona, Spain

**Keywords:** physical activity, sedentary behavior, type 2 diabetes mellitus, healthy behavior, healthcare professionals, primary healthcare settings

## Abstract

Maintaining an active lifestyle is a key health behavior in people with type 2 diabetes (T2D). This study assessed the feasibility and acceptability of a socio-ecological Nordic walking intervention (SENWI) to enhance healthy behaviors in primary healthcare settings. Participants included individuals with T2D (*n* = 33; age 70 (95% CI 69–74)) and healthcare professionals (HCPs, *n* = 3). T2D participants were randomly assigned to a SENWI, active comparator, or control group for twelve weeks. Feasibility and acceptability were evaluated based on a mixed methodology. Quantitative data reported adherence information, differences between follow-up and dropout participants and pre- and post-intervention on physical activity, sedentary behavior, and health outcomes. Qualitative data acquisition was performed using focus groups and semi-structured interviews and analyzed using thematic analysis. Thirty-three T2D invited participants were recruited, and twenty-two (66.7%) provided post-intervention data. The SENWI was deemed acceptable and feasible, but participants highlighted the need to improve options, group schedules, gender inequities, and the intervention’s expiration date. Healthcare professionals expressed a lack of institutional support and resources. Nevertheless, no significant difference between the SENWI follow-up and dropout participants or pre- and post- intervention was found (only between the active comparator and control group in the physical quality of life domain). Implementing the SENWI in primary healthcare settings is feasible and acceptable in real-world conditions. However, a larger sample is needed to assess the program’s effectiveness in improving healthy behaviors and its impact on health-related outcomes in the long term.

## 1. Introduction

Type 2 diabetes mellitus (T2D) is the most common metabolic disease and a leading cause of mortality worldwide [1]. It has been estimated that more than 462 million people (6.28% of the world population) are currently living with T2D [2]. T2D increases in prevalence throughout the human lifespan, and because of that, it is expected to increase in the coming decades due to population ageing [2].

The cost attributable to T2D increases substantially for older age groups and for those with micro- and macrovascular complications [3]. In Spain, the annual average healthcare cost to treat T2D complications is EUR 3110.10, which is 72.4% higher than non-diabetics [4]. The attributable costs are primarily incurred through acute inpatient hospitalizations, physician visits, medication, and assistive devices [3]. The excess healthcare costs attributable to T2D pose a significant clinical and public health challenge [3].

The cornerstones of T2D treatment are medication, diet, and lifestyle, such as reducing and limiting the time spent in sedentary behaviors (SBs—defined as any waking behavior in a sitting or reclining posture costing ≤1.5 times the basal metabolic rate) [5] and increasing physical activity (PA) [6]. The benefits of PA have been extensively studied, including improvements in blood glucose levels [7], quality of life [8], life expectancy, and reductions in complications [6,9], as well as the cost burden associated with T2D complications [10]. Nevertheless, only 9% of the population living with T2D met the recommendations of the World Health Organization (WHO) (that is, 150 min moderate-intensity or 75 min vigorous-intensity PA per week) [11]. 

Evidence has reported effective results when meeting the WHO recommendations with a wide variety of PA programs, such as active living interventions that promote less sitting and more movement in people with T2D [12] or exercise prescription programs such as high-intensity interval training [13,14,15] or walking interventions, including Nordic walking [16,17,18]. However, adherence in real-world situations to PA programs remains a social problem, particularly among individuals with chronic diseases such as T2D [19].

Adopting and maintaining active behaviors involves a multiple-interaction approach: (i) patient-related factors, (ii) health system factors, (iii) condition-related factors, (iv) therapy-related factors, and (v) socioeconomic factors [20]. Studies have indicated that socioeconomic status [21], gender [22], chronic degenerative disease, age discrimination [23], and other structural factors act as barriers to maintaining healthy behaviors. In this regard, T2D prevalence increases with age; people with T2D are usually elderly, with all the barriers associated with this part of the population [21,23]. Thus, while focusing on a single approach may enhance short-term and individual-level changes, multiple approaches may facilitate long-term population-level effects in real-world situations [24]. To cope with this complex situation, interventions based on a socio-ecological model may gather better results to improve adherence to healthy behaviors, including PA [21,25,26,27].

Primary healthcare is crucial for tackling the complexities of adhering to healthy behaviors and the diverse factors that influence individuals with T2D [28]. However, primary healthcare attention mainly focuses on providing general advice on healthy behaviors and PA guidelines without considering the other factors that may influence adherence [29]. Nevertheless, healthcare professionals (HCPs) often feel that they have limited time and resources, a lack of knowledge, and a lack of options to support patients’ adherence to PA [28,30]. As a result, HCPs often achieve poor results in changing users’ long-term health behaviors and PA, which may discourage them from trying to promote them [31], despite their potential to improve glucose control [32,33] and promote healthy behaviors in the population, especially among people with chronic diseases [28].

Given the scale and span of the benefits for practicing regular and sustainable PA in people who live with T2D, it is essential to design and evaluate socio-ecological interventions in primary care base settings that improve PA adherence as a treatment for T2D. Therefore, the purpose of this study was twofold: (i) to assess the changes in PA, SB, metabolic and health outcomes, and quality of life from pre- to post-program and the differences between follow-up and withdraw participants, and (ii) to assess the feasibility and acceptability of the SENWI program to improve adherence in primary-care-based people with T2D.

## 2. Materials and Methods

### 2.1. Study Design

This is a pragmatic randomized controlled trial with three groups (SENWI, active comparator, and control) of recruited people with a diagnosis of T2D (NCT05159089). The methodology of the study has been described elsewhere and was published as a protocol [34]. Briefly, a mixed-method methodology was used: first, a quantitative approach was used to assess the effects from pre- to post-intervention and the differences between follow-up and withdraw participants, and second, a qualitative approach was used to assess the acceptability and feasibility of the intervention.

### 2.2. Participants

Participants were recruited from the Monistrol de Montserrat primary care setting (Barcelona) between January and March 2022. During routine practice visits, a T2D nurse specialist invited people with T2D who met the eligibility criteria to voluntarily participate in the study, using a random sample. The inclusion criteria for eligible participants were (i) having T2D for more than 2 years, (ii) aged between 60 and 80 years; and (iii) possessing no major physical limitations prescribed by the doctor or any HCP. Participants were excluded if they (i) were unable to provide informed consent; (ii) were unable to understand the study materials and instructions due to mental illness; (iii) had complications such as neuropathy, retinopathy, and nephropathy; (iv) had relative or absolute contraindications to perform PA; and (v) had a body mass index (BMI) over 34.9 kg/m^2^ since Nordic walking (NW) requires free arm mobility that is impaired by a body mass index over 40 kg/m^2^. 

The T2D nurse specialist invited to participate 47 people who fulfilled the inclusion criteria. From those, 33 agreed to participate in the study and signed the informed consent voluntarily. Reasons for not participating were freely given when participants declined the invitation and included (i) not being interested in and (ii) not wanting to participate in group activities (see Figure 1). They were scheduled to attend the first round of data collection by the research team in April 2022. Confidentiality of personal information was ensured in accordance with the Protection of Personal Data, the Guarantee of Digital Rights, and the General Regulation (EU) 2016/679 of 27 April 2016. Ethical approval was granted by the Research Ethics Committee of the University of Vic, Central University of Catalonia (2021), and the IDIAP Jordi Gol Ethics Committee (2022).

### 2.3. Randomization and Blinding

A computerized random number generator (http://www.randomization.com, accessed on 11 January 2022, created by Dr Gerard E. Dallal, Tufts University; accessed on 11 January 2022) was used to randomize eligible participants at the level of the individual participant, stratified by sex and age, in blocks of five. Concealed randomization was conducted by a research member after each participant had been included in the study, assigned an identification code, and completed the study baseline assessment. Allocation concealment was ensured, as the random number generator was not released until the participant was recruited into the trial. The participants were notified personally face-to-face.

The trial had an open-label design with a blinded assessment of the outcomes. The researchers and participants carrying out the baseline assessments were blinded to the group allocation. The statistician was also blinded to the group allocation until the statistical analysis was completed. Participants were asked not to reveal their group allocation when undergoing follow-up measurements. To assess the extent to which blinding had been preserved, researchers recorded the number of cases in which the allocation was revealed.

### 2.4. Study Interventions

A three-arm randomized controlled trial was conducted between May and July 2022. Participants were informed that participation in a PA activity group (NW, and socio-ecological Nordic walking intervention—SENWI) was required 2 times/week for 12 weeks. NW provides greater health benefits (e.g., glucose control, cardio-respiratory fitness, flexibility, and upper-body strength) compared with merely walking [16,18], and it is easier to perform than high-intensity interval training [35].

Participants in the control group were informed that they would be required to increase PA and break up long periods of SB freely and that different outcomes should be registered before and at 12 weeks for the health and wellbeing of the healthcare-setting users. Before the intervention started, the SENWI and active comparator groups had undertaken a beginner’s class in NW over two days with an instructor (RDR and SNG) at the study site. After these two sessions, a 12-week intervention with 2 sessions per week was conducted. The meeting point was in the primary healthcare setting. The SENWI group performed NW similar to the active comparator group, but the instructors were instructed to use a socio-ecological approach based on the participants’ awareness of social determinants of health and PA with adaptations for people living with T2D [34,36,37].

All instructors in charge of conducting the SENWI and active comparator group were trained by the research team to apply a standardized protocol. The methods for delivering both interventions have been previously described [34]. Both intervention (SENWI and active comparator) sessions took place in the green areas surrounding the primary healthcare setting of the Monistrol de Montserrat. The sessions were conducted on a flat track with a maximum cumulative altitude of 100 m.

All participants continued with their pharmacological treatment plans and the nutritional and PA recommendations provided by their HCP treatment. Moreover, all three groups were expected to have no harmful effects, as they only involved low- and moderate-intensity PA. A protocol that gradually increased time and intensity was implemented to avoid patient discomfort due to PA.

### 2.5. Measures, Data Collection, and Management

#### 2.5.1. Quantitative Assessments

All researchers in charge of conducting the assessment (GJC and SNG) undertook a standardized training session. Quantitative assessments were conducted by the research team at baseline, pre-, and post-intervention. All assessments were conducted in the Monistrol de Montserrat primary healthcare setting.

Demographic outcomes included socioeconomic status and demographic and background information including sex, age, marital status, family status, obligations, and medication.

Quantitative outcomes included (i) program dropout ratio, (ii) light-intensity PA (LPA) and moderate to vigorous PA (MVPA), (iii) weekly steps and stepping time, (iv) sitting and standing time, (v) sitting bouts and transitions from sitting to standing (breaks), (vi) metabolic and health outcomes (HbA1c, waist circumference, BMI, and quality of life). The same nurse that recruited the participants collected all data except for activPAL^TM^ (outcomes ii, iii, iv, and v), which were collected by the research members (GJC and SNG). Qualitative outcomes were collected by research members (GJC and SNG) and included the feasibility and acceptability program. All outcome measurement methods have been described previously [34].

To assess safety and well-being during the intervention period, patients’ perceptions about health, injury, pain, and intervention were asked every week by a specially trained professional physiotherapist in charge of delivering the NW (SNG).

The confidentiality of personal information about potential and enrolled participants was protected using codification (i.e., number of participants, number of groups, number of outcomes assessment) before, during, and after the trial by the principal investigator (GJC) on a server located at the University of Vic—Central University of Catalonia (i.e., Faculty of Health Science at Manresa).

#### 2.5.2. Quantitative Data Analysis

Descriptive statistics were reported as medians and interquartile (Md_95%CI) or frequencies (*n*, %), as appropriate. To evaluate the differences between withdrawn and follow-up participants, a Mann–Whitney’s test was used to compare demographic characteristics and primary outcomes recorded at the baseline for all samples and for specific groups. Although hypothesis testing was not adequate for primary outcomes because the study was not adequately powered to detect statistically significant differences between groups, a Kruskal–Wallis test was used to compare groups at the baseline and post-intervention.

#### 2.5.3. Qualitative Assessments

Feasibility and acceptability were assessed by focus groups with participants and semi-structured interviews with HCPs between July and June 2022. The focus groups of the intervention were conducted by two research team members (GJC and SNG) with (i) participants with T2D who finished the intervention and (ii) participants who dropped out of the intervention. Semi-structured interviews were used to collect data from HCPs that had to implement NW and the SENWI.

The focus groups had four to six participants and lasted between 45 and 60 min, while the semi-structured interviews lasted around 60 min. Before any audio recording was conducted, participants were reminded of the aims and invited to ask any questions they had. The sessions were conducted in Spanish in primary healthcare settings. No judgments, criticisms, or (dis)approval of contributions were expressed during the sessions. Participants in each focus group were allocated based on their dropout rates.

The focus groups and semi-structured interviews explored various topics related to the intervention, such as barriers, enabling follow-up, feasibility (applicability and improvements on the program), acceptability (reasons to withdraw or participate), and potential changes to improve feasibility and adherence. The guide questions were revised after each session to explore themes emerging from the previous groups. Prior to formal data collection, the focus group process was performed following the Consolidated Criteria for Reporting Qualitative Research (COREQ) guidelines [38].

#### 2.5.4. Qualitative Data Analysis

A trained moderator (GJC) led and conducted focus groups and semi-structured interview sessions using a guide with open-ended questions to explore the study’s aims [39,40]. A moderator’s assistant (SNG) took notes from which a summary was provided to participants for verification at the end of the session. Besides the participants and researchers, there was no one else present during the sessions. Participants’ responses were audio recorded and then fully transcribed and subjected to a series of five iterative steps to conduct a descriptive inductive thematic analysis relating to “PA program feasibility” using Atlas.ti [41]: (i) familiarization with the data; (ii) inductive open coding to generate initial codes; (iii) searching for emerging themes within the codes generated from the patient and HCP transcripts; (iv) reviewing codes and themes; and (v) identifying the key factors which influence sitting less and moving more in the setting. The final step was based on two criteria: (i) the importance expressed by participants (repetition and depth of discussion during sessions) and (ii) repetition in the different focus groups/semi-structured interviews. The most important factors were those that appeared more often, repeatedly, and with more depth of expression in the participants [40]. Two researchers (GJC and SNG) independently performed the codification and thematic analysis, and then discussed and agreed on the key themes and data saturation. The themes were derived from the data. Transcripts and themes were returned to the participants for verification and feedback on the findings. Participants’ quotes to support themes were identified and translated from Catalan to English.

## 3. Results

### 3.1. Participants Characteristics

Thirty-tree invited participants (70%) were enrolled in the study. The median age was 70 years old (95% CI 69–74), two-thirds were male (63.6%), and 21 reported that they were physically active (APAFB questionnaire). Nineteen participants (57.6%) had been diagnosed with T2D more than 10 years ago, 66.7% had a low socioeconomic level, the median BMI was 30.7 (95% CI 29.1–33.1), and the median waist circumference was 109 (95% CI 104.9–112.3) (see Table 1). The number of steps medium was 7909.1 (95% CI 6679.9–10747) with a walking duration of 107.1 (95% CI 90.6–129.6) minutes. The median number of total breaks of SB (i.e., sit-to-stand transitions) was 45.5 (95% CI 43–49.7), and the absolute time spent in SB was 564.3 (95% CI 531–598.2) minutes per day. The LPA medium time was 90.9 (95% CI 81.9–107.2) minutes, and the MVPA time was 9.6 (95% CI 6.7–17.3) minutes. All PA and SB outcomes at the baseline are shown in Table 1. At the baseline, only a significant difference between the SENWI and control groups was found in light PA (*p* = 0.031) (see Table 1).

### 3.2. Interventions Impact on PA, SB and Health Outcomes

Participants only mentioned muscle soreness during the first weeks of intervention. No other safety or well-being problems during the intervention were mentioned. The pre- and post-intervention primary outcomes within the groups are displayed in Table 2. Only a significant difference between the NW and control groups was found in SF-12 (physical dimension) (*p* = 0.028) post-intervention. No other significant differences were observed between the groups post-intervention.

### 3.3. Intervention Adherence and Withdraw

During the study, four participants dropped out in the first month, two in the second month, and five in the third month (*n* = 11). The reasons for dropping out of the study were health issues unrelated to PA (*n* = 7) or the intervention and change in residence (*n* = 4) (see Figure 1). The adherence to interventions (SENWI and NW) for those who finished it (*n* = 22) was 71.41%. The SENWI group presented 68.88% adherence in comparison to the 72.83% adherence in the NW group (*p* = 0.518). Women in the SENWI group showed less adherence than men (67.7% vs. 73.4%) but with no statistically significant difference (Mann–Whitney test: *p* = 0.767). The main reason for not attending the sessions (i.e., adherence) was because of a positive COVID-19 test (24.2%), non-negotiable family responsibilities (i.e., taking care of the grandchildren or partner) (57.3%), and having a medical visit the same day or health issues (18.5%). No adverse events due to the intervention have been registered. Two scheduled days were suspended because of climate issues (one because it was raining and the other because of a fire near the primary healthcare setting).

No significant statistical differences were found between the withdrawal and follow-up participants at baseline for all samples (see Table 3). No statistical differences were found between the follow-up and withdrawal of participants when we conducted the analysis for all samples (see Table 2) or for each group. In addition, there was no association between group allocation and the risk of withdrawal from the study (*p* = 0.379) (see Table 3).

### 3.4. Intervention Feasibility and Acceptability

All participants with T2D from the SENWI group (*n* = 10) were intentionally selected for inclusion in two focus groups (one with follow-up and another with withdrawn participants). All HCPs involved in the study were also intentionally selected for inclusion in three semi-structured interviews. All participants with T2D expressed that the intervention was enjoyable; however, participants also provided input to improve and increase the feasibility and acceptability of the SENWI.

#### 3.4.1. People with T2D Perceptions

Descriptive inductive thematic analysis of the open-ended questions revealed three themes and seven subthemes between the T2D participants:Theme 1: Useful but with expiry date

Participants expressed that although the use of poles and the changes in NW routes helped them to be more adherent to the PA intervention, the program was not enough for them to keep going once the intervention was completed.

“*At first, I had my doubts, but then, when I try the poles, they were helpful. And not only at the uphill, but also downhills!*”(man; at the follow-up of the intervention)

Moreover, participants indicated that the program provided a good spark to change their healthy behaviors (i.e., adopt); however, they expressed the need for reminded activities to keep going with these changes in the long run.

“*It was good to start doing something new, but once it is finish, what? What we have to do now to keep going on?*”(man; at the follow-up of the intervention)

Theme 2: Gender inequities

Participants had different reasons for attending sessions related to gender inequities. Women showed attendance barriers, such as kinship needs, that were seldom referred to by men.

“*Sometimes I had to choose to come here or to take care of my grandchildren. Sometimes I was not able always to skip my family duties.*”(woman; at the follow-up of the intervention)

Otherwise, male participants referred to a good program because it efficiently fulfilled their daily free time. This was seldom referred to by women, who viewed participation more as an obligation in light of schedule problems.

“*It was great. I had a lot of free time, which was a good way to stay occupied throughout the day.*”(man; at the follow-up of the intervention)

Theme 3: Different PA capacity between individuals

When participants were asked questions about what changes they thought might improve the feasibility of the intervention, the main response was the heterogeneity of PA capacity between participants.

“*For me it was sometimes boring because other participants walk too slow.*”(man; dropout of the intervention)

“*It was not easy for me. It takes me a lot of time and get bored, that is the reason I drop out.*”(woman; dropout of the intervention)

“*I think it will be better to know what capacities have each one before we are assigned to a group.*”(woman; at the follow-up of the intervention)

#### 3.4.2. HCP Perceptions

A descriptive thematic analysis of the open-ended questions revealed two themes and five subthemes between HCPs:Theme 4: Feasibility in the healthcare system

When HCPs were asked to question the feasibility, acceptability, barriers, and that which would enable the implementation of the intervention in a healthcare-setting system, the responses focused on the lack of resources and insufficient institutional support to keep the intervention going once it is finished.

“*It was easy while the study was going on. But once it is finish; it is impossible to carry on without any help.*”(HCP; nurse specialist)

“*Without more resources and time, or even more, more healthcare professionals to do this job, I think that is not possible nowadays.*”(HCP; physiotherapist)

Theme 5: Intervention with bounded options

Based on the responses to the question of what changes in the intervention may improve adherence to it, HCPs responded that they lacked a greater variability in the schedule options for participants and the possibility of creating more groups related to the different capabilities of the participants (e.g., low- and moderate-intensity groups).

“*At the beginning it was difficult to enroll some participants because they were not able to enroll in the morning groups. Maybe with more schedule groups will be easy.*”(HCP; nurse specialist)

“*It was clear from the beginning that some participants had difficulties to follow-up the sessions, while others express that we were going to slow. I think that some of them left the study because of that.*”(HCP; physiotherapist)

## 4. Discussion

This study assessed (i) the feasibility and acceptability of a PA socio-ecological approach (SENWI) in primary-care-based people with T2D and (ii) changes in PA, SB, and metabolic and health outcomes from pre- to post-program and between participants who withdrew or followed up the study. Feasibility and acceptability findings provided positive feedback of the SENWI at both the individual (i.e., participants) and context (i.e., HCPs involved) levels. At post-intervention, only a significant difference in the physical dimension (SF-12) was found between the NW and control groups. No other statistically significant differences were observed between groups at post-intervention.

The main reason for dropout was health issues. To reduce this dropout related to health issues, studies have suggested improving health literacy [42]. Although the SENWI program takes into consideration different dimensions of healthy behaviors (including healthy literacy), we did not find differences between the groups in dropout ratio. Another main reason for withdrawal was a change in residence. Although this is difficult to control, universalizing at the national level this kind of PA intervention in primary healthcare systems may be a good solution [43]. In Catalonia (Spain), the recent introduction of a physiotherapist in healthcare settings could help develop this kind of intervention nationwide.

For those completing the SENWI program, adherence was medium-to-high (68.88%) and focus groups and interviews reported positive responses in the qualitative study evaluation of feasibility and acceptability. All follow-up participants enjoyed the intervention and referred to poles as being helpful for pushing while walking, but some had problems following the group (i.e., different PA capacity between individuals). Moreover, participants viewed the SENWI as a good spark to start and express the necessity to permanently implement this kind of intervention in the primary healthcare system. Previous feasibility studies have reported similar results when implementing PA interventions [44]. However, this was not enough to change their health behaviors in the long run (i.e., useful but with an expiry date). For example, women referred to problems in participating in the SENWI sessions due to kinship needs (i.e., gender inequities). Indeed, although not statistically significant, women who completed the SENWI program had lower adherence rates than men. This gender inequity in PA access had already been exposed: women and girls are used to having more barriers to access and adhere to PA because it is seen as a masculine pursuit [29] and because they are used to holding responsibility for kinship/family needs [45]. Furthermore, age discrimination also influences physical activity (PA) capabilities, especially in people with type 2 diabetes (T2D) [29], who are commonly elderly. Participants often mention the difference between their PA abilities when participating in the program. Some participants experience difficulties with the intensity, despite it being low to moderate, while others feel that they are going too slowly. In this sense, PA adherence is often impeded by dynamic, unstable, and unpredictable recurrent socioeconomic conditions that may often overwhelm the personal capability to keep performing PA [29]. Although these social inequity barriers were not related to the reasons for withdrawing, they may affect PA and sedentary behavior and, therefore, health outcomes in the long run. Thus, these contextual social issues present a near-universal challenge for adhering to PA.

Finally, the HCPs expressed that the SENWI feasibility was limited due to insufficient institutional support and the lack of resources to implement the intervention (i.e., intervention with bounded options). Therefore, it is necessary to integrate PA interventions efficiently and permanently in primary healthcare settings through a cooperative planning process (i.e., feasibility in the healthcare system) [46]. The cooperative planning process focused on the participation of different stakeholders: the target group, physical educators and physiotherapists, policymakers, and researchers [46,47]. Moreover, it takes into consideration factors that affect the adoption and maintenance of active behaviors, which include the social and economic dimension (e.g., social inequity), the personal (e.g., gender and age discrimination), illness, and treatment dimensions, such as healthcare professionals (HCPs), and the healthcare system (e.g., accessibility) [21,48]. This may be a good solution to implementing the SENWI program, obtaining institutional and policymakers’ support, and ending users’ needs to reduce their social inequities once research interventions are completed. Otherwise, when primary healthcare attention is deficient due to a lack of resources or accessibility (i.e., rural areas), any improvement in quality of life and health decreases significantly in people with T2D [33].

### Strength and Limitations

This study is unique, as it is the first to implement a socio-ecological PA intervention in primary healthcare based on people with T2D. Moreover, this approach co-creates an intervention throughout the research to be used systematically in real-world situations [49]. Thus, the strengths of this study include capturing data at both the individual and setting levels, the dissemination of an intervention in a real-world setting, and the use of an active comparator with NW that allows the isolation of the socio-ecological effects in the SENWI. Another strength of this study is the use of a socio-ecological approach to guide implementation [50], taking into consideration social factors that may affect participation, adherence, and dropout in PA interventions [51,52,53].

The trial had an open-label design due to the nature of the research where participants were unable to be blind to group allocation. While our small sample size and attrition rate provided acceptability and feasibility information, it limited our ability to conduct between-group statistical analyses and draw conclusions about effectiveness. Based on the nature of the study, we used mediums and confidence intervals with non-parametric analysis (Mann–Whitney and Kruskal–Wallis tests) to obtain a better representation of the small sample. Future studies should take into consideration these limitations. Thus, more studies are required to assess the effectiveness of the SENWI on healthy behaviors and outcomes. Another limitation was the emergence of the COVID-19 pandemic during the study recruitment and implementation phase (December 2021–July 2022), affecting the participants’ participation and adherence to the program.

## 5. Conclusions

After assessing the feasibility and acceptability throughout a qualitative analysis, the findings of this study highlight that existing real-world T2D healthcare should consider incorporating permanent PA interventions with a socio-ecological approach (e.g., SENWI) to support long-term habitual PA in people with T2D. No important significant differences were observed between the post-intervention groups (only in the SF-12 physical dimension between the NW and control group). Although this study presents a step forward in closing this research-to-practice gap, future research should investigate the effects of this intervention or efficiently incorporate similar interventions into primary healthcare settings permanently.

### Future Implications

To implement PA interventions throughout primary healthcare settings it may be necessary to understand the social inequalities between users (gender, age, or socioeconomic inequities). It may also be necessary to conduct PA interventions through socialization, without any cost, and permanently throughout primary healthcare settings. In this sense, future studies should adopt a pragmatic design, such as a cooperative planning process, to permanently integrate PA study interventions into real-world situations [44,46]. Without policymakers, HCPs, or target group support, PA interventions will fail, especially once the intervention studies finish.

## Figures and Tables

**Figure 1 healthcare-11-01815-f001:**
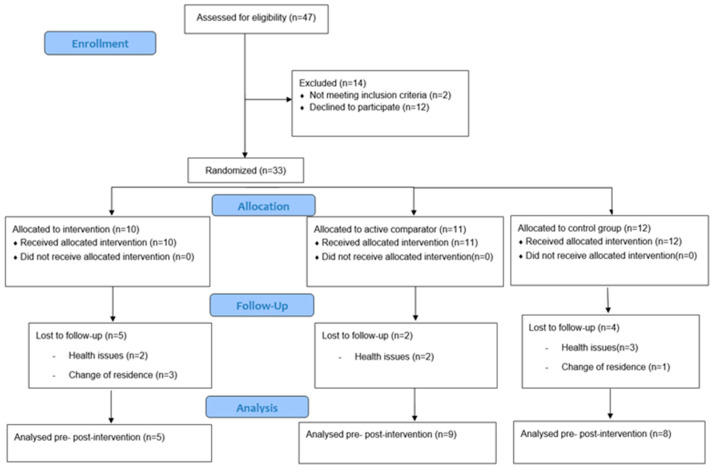
CONSORT diagram of the flow of participants through the study, including reasons for declining participants and withdrawment.

**Table 1 healthcare-11-01815-t001:** Characteristics of study participants at baseline.

Participant Characteristics at Baseline	SENWI Group (n = 10)	NW Group (n = 11)	Control Group (n = 12)	Full Sample (n = 33)
	*n* (%)	*n* (%)	*n* (%)	*n* (%)
Sex				
Female	4 (40)	3 (27.3)	5 (41.7)	12 (36.4)
Male	6 (60)	8 (72.7)	7 (58.3)	21 (63.6)
Years from diagnosis				
>2 and <5	3 (30)	3 (27.3)	1 (8.3)	7 (21.2)
>5 and <10	1 (10)	5 (45.5)	1 (8.3)	7 (21.2)
<10	6 (60)	3 (27.3)	10 (83.3)	19 (57.9)
Socioeconomic status				
Low	7 (70)	8 (72.7)	7 (58.3)	22 (66.7)
Medium	0 (0)	2 (18.2)	2 (16.7)	4 (12.1)
High	3 (30)	1 (9.1)	3 (25)	7 (21.2)
Physically Active (APAFB)				
No	3 (30)	6 (54.5)	3 (25)	12 (36.4)
Yes	7 (70)	5 (45.5)	9 (75)	21 (63.6)
Participant characteristics at baseline	SENWI Group (*n* = 10)	NW Group (*n* = 11)	Control Group (*n* = 12)	Full sample (*n* = 33)
	Md (CI 95%)	Md (CI 95%)	Md (CI 95%)	Md (CI 95%)
Age (years)	69.5 (67–74)	69 (67–73)	72.5 (66–78)	70 (69–74)
Waist circumference (cm)	108 (95.9–113.8)	109 (101.7–114.6)	111 (96.5–118.8)	109 (104.9–112.3)
BMI (kg/m^2^)	30.3 (25.3–33.7)	33.1 (27.8–35.1)	30.1 (26.8–36.4)	30.7 (29.1–33.1)
HbA1c (%)	6.5 (6.2–7.5)	7.1 (6.2–7.7)	6.7 (5.9–7.4)	6.8 (6.4–7.1)
SF-12 (PCS)	48.6 (28.7–53.5)	48.4 (41.9–54.5)	34.2 (27.2–51.5)	44.3 (39.4–48.9)
SF-12 (MCS)	58.2 (44.8–61.9)	54.5 (45.1–54.5)	44.7 (34.8–56.5)	54.4 (45.1–57.3)
Steps/day	12,732.1 (7392.7–15,587.3)	7239 (5033.2–9407.7)	7536.6 (4948.1–10,918.8)	7909.1 (6679.9–10,747)
Walking duration (min)	150 (93.6–190.2)	96 (72.6–126)	97.5 (75–140.4)	107.1 (90.6–129.6)
Stand-up duration (min)	291 (201–313.2)	226.8 (169.2–301.8)	224.1 (183.6–252.6)	231.9 (207.6–253.2)
Absolute time in SB (min)	560.4 (413.4–615.6)	595.8 (491.4–621)	564.3 (517.2–706.8)	564.3 (531–598.2)
Bouts < 30 min (min)	247.5 (203.7–375.1)	220.9 (189.8–292.3)	298.7 (211.1–416.6)	247.5 (220.1–301.8)
Bouts between 30 and 60 min (min)	145.71 (81.9–166.6)	126.5 (99.7–147.4)	142.6 (61.8–161.1)	138.7 (107.8–151.9)
Bouts > 60 min (min)	107.7 (63.3–217.5)	182.9 (128.6–231.1)	125.7 (65.6–288.9)	156.9 (115.8–205.9)
Sit-to-stand transitions (number)	43 (37.1–55.8)	45 (35.7–48.9)	46.5 (43.3–73.3)	45.5 (43–49.7)
Number of SB bouts < 30 min	40 (32.2–49)	37 (29.7–43.3)	41.5 (39.3–70)	41 (37.3–43.9)
Number of SB bouts between 30 and 60 min	3 (2–4)	3 (2–4)	3 (1.3–4)	3 (2.1–4)
Number of SB bouts > 60 min	1 (1–2)	2 (1.7–3)	1.5 (1–2.7)	2 (1–2)
LPA duration (min)	109.6 (89.7–125.6) *	89.1 (66.9–111.2)	80.6 (63.6–98.3) *	90.9 (81.9–107.2)
MVPA duration (min)	20.9 (3.8–64.7)	8.9 (6.8–17.3)	8.3 (0.7–49.3)	9.6 (6.7–17.3)

*p*-value *: Kruskal–Wallis *p*-value = 0.031 (post-hoc analysis Mann–Whitney test showed a significant difference between the SENWI and control groups (*p*-value = 0.021)). Md: medium; CI: confidence interval; *n*: number; %: percentage; SENWI: socioecological Nordic walking intervention; NW: Nordic walking intervention; APAFB: brief physical activity questionnaire; BMI: body mass index; HbA1c: glycated hemoglobin; SF-12: quality of life questionnaire; PCS: physical dimension; MCS: mental dimension; SB: sedentary behavior; LPA: light physical activity; MVPA: moderate–vigorous physical activity.

**Table 2 healthcare-11-01815-t002:** Primary metabolic and physical-activity outcome comparison between pre- and post-intervention.

Participant Primary Metabolic and PA Outcomes	SENWI Group (n = 5)		NW Group (n = 9)		Control Group (n = 8)	
	Pre	Post	Pre	Post	Pre	Post
	Md (CI 95%)	Md (CI 95%)	Md (IC 95%)	Md (CI 95%)	Md (CI 95%)	Md (CI 95%)
Waist circumference (cm)	108 (89–114)	105 (86–111)	109 (69.9–113.7)	109 (100.3–113.9)	107 (95–122.2)	104.5 (93.1–119.3)
BMI (kg/m^2^)	31.1 (24.2–34.3)	31.7 (24.1–32.1)	33.1 (25.1–35.5)	32.8 (24.7–35.2)	29.4 (24.8–38.5)	28.1 (24.8–37.9)
HbA1c (%)	6.4 (5.8–7.6)	6.8 (6.1–8)	7.1 (6.1–8.7)	6.9 (6.3–7.8)	6.9 (5.8–7.5)	6.9 (6.2–8.5)
SF-12 (PCS)	44.3 (26.2–53.8)	50.8 (27.7–53.4)	48.4 (40.5–53.5)	49.7 (42.4–52.4) *	33.8 (26.2–54.9)	30.9 (24.1–48.5) *
SF-12 (MCS)	58.9 (33.2–62.2)	59.9 (41.4–64.7)	54.5 (50.3–60.7)	57.9 (51.1–59.1)	42.7 (31.6–59.8)	42.1 (27.9–64.4)
Steps/day	10,146.6 (6299–17,007)	8769.5 (7321–10,404)	7501 (5786.3–10,642.7)	6921 (5913.3–10,509.1)	7536.6 (3326.4–13,974.4)	8142 (3225.7–18,870.9)
Walking duration (min)	127.8 (90–194.4)	119.4 (88.8–130.8)	99.6 (78.6–143.6)	99 (79.8–141.6)	97.5 (53.8–171.6)	112.2 (47.4–196.2)
Stand up duration (min)	261.3 (228.6–301.8)	270 (205.1–317.8)	236.4 (180.9–321.6)	254.4 (147.6–341.4)	213.3 (121.2–268.7)	242.4 (196.8–337.8)
Absolute time in SB (min)	561 (333–616.8)	598.8 (514.2–656.4)	595.8 (445.4–616.4)	545.4 (456–660)	552.9 (482.4–739.2)	543 (430.2–686.4)
Bouts < 30 min (min)	246.3 (171.5–377.1)	244.4 (220.7–407.1)	220.9 (191.6–287.8)	253.6 (199.4–338.7)	277.6 (138.3–445.3)	260.5 (181.1–369.2)
Bouts between 30 and 60 min (min)	121.4 (75.7–166.7)	152.6 (92.3–196.2)	126.5 (96.1–147.1)	125.3 (104.8–202.9)	123.9 (41.9–157.5)	151.9 (80.8–178.1)
Bouts > 60 min (min)	126.3 (71.3–221.8)	180.7 (84.2–211.5)	164.4 (125.9–225.2)	134.9 (94.1–217.7)	165.8 (38.9–329.9)	102.9 (53.3–282.9)
Sit-to-stand transitions (number)	43.5 (35–67)	49 (38–68)	45 (36.2–47.9)	49 (37.6–57.1)	46.5 (30.6–85.3)	39 (31.6–63.1)
Number of SB bouts < 30 min	39 (30–64)	44 (30–64)	37 (30.1–42.9)	45 (31.6–50.8)	41.5 (26.9–82.6)	34 (26.3–58.4)
Number of SB bouts between 30 and 60 min	3 (2–4)	3.5 (2–5)	3 (2–3.9)	3 (2–5)	2.5 (0.7–4)	4 (1.6–4)
Number of SB bouts > 60 min	1.5 (1–2)	2 (1–2)	2 (1.8–2.9)	2 (1–2)	2 (0.7–3)	1 (1–3)
LPA duration (min)	104.3 (81.9–126.1)	105.4 (81.2–107.5)	92.8 (74.3–135.9)	90.6 (64.1–131.77)	104.3 (52.3–114.1)	87.6 (37.8–111.9)
MVPA duration (min)	25.2 (4.1–68.6)	13.9 (6.6–24.1)	11.2 (6.9–17.3)	10.8 (3.4–18.9)	25.2 (5.1–62.1)	7.5 (1.5–113.1)

*p*-value *: Kruskal–Wallis *p*-value = 0.028 (post-hoc analysis Mann–Whitney test showed a significant difference between the Nordic walking and control groups (*p*-value = 0.012)). Md: medium; CI: confidence interval; SENWI: socio-ecological Nordic walking intervention; NW: Nordic walking intervention; BMI: body mass index; HbA1c: glycated hemoglobin; SF-12: quality of life questionnaire; PCS: physical dimension; MCS: mental dimension; SB: sedentary behavior; LPA: light physical activity; MVPA: moderate–vigorous physical activity.

**Table 3 healthcare-11-01815-t003:** Comparison of the characteristics at baseline between follow-up and withdraw participants.

Participant Characteristics at Baseline	Follow-Up (n = 22)	Withdraw (n = 11)
	n (%)	n (%)
Sex		
Female	9 (40.9)	3 (27.3)
Male	13 (59.1)	8 (72.7)
Years from diagnosis		
>2 and <5	3 (13.6)	4 (36.4)
>5 and <10	5 (22.7)	2 (18.2)
<10	14 (63.6)	5 (45.5)
Socioeconomic status		
Low	13 (59.1)	9 (81.8)
Medium	3 (13.6)	1 (9.1)
High	6 (27.3)	1 (9.1)
Physically Active (APAFB)		
No	9 (40.9)	3 (27.3)
Yes	13 (59.1)	8 (72.7)
Group		
SENWI	5 (50)	5 (50)
NW	9 (81.8)	2 (18.2)
Control	8 (66.7)	4 (33.3)
Participant characteristics at baseline	Follow-up (*n* = 22)	Withdraw (*n* = 11)
	Md (CI 95%)	Md (CI 95%)
Age (years)	69.5 (67–74)	71 (69–74)
Waist circumference (cm)	108.5 (100.6–111.2)	111 (103.4–117.7)
BMI (kg/m^2^)	30.8 (27.3–33.1)	30.7 (27.6–35.1)
HbA1c (%)	6.7 (6.2–7.2)	6.9 (6.2–7.5)
SF-12 (PCS)	43.5 (34.9–49.1)	48.8 (32.6–53.7)
SF-12 (MCS)	54.4 (44.4–59.1)	55.7 (43.8–59.6)
Steps/day	7531.1 (6438.7–9914.6)	10,430 (5039.5–14,406.5)
Walking duration (min)	99.6 (88.2–133.2)	121.2 (72.6–180.6)
Stand-up duration (min)	231.6 (208.8–268.8)	250.2 (166.8–313.2)
Absolute time in SB (min)	567.6 (519–608.4)	561 (448.8–635.4)
Bouts < 30 min (min)	236.9 (210.1–298.7)	289.7 (203.7–348.1)
Bouts between 30 and 60 min (min)	126.5 (88.1–150.3)	145.7 (119.5–175.3)
Bouts > 60 min (min)	164.4 (106.6–215.5)	127.6 (66.1–236.5)
Sit-to-stand transitions (number)	46 (38.9–49.1)	44 (38.4–54.9)
Number of SB bouts < 30 min	41 (34–43.5)	40 (32.5–49)
Number of SB bouts between 30 and 60 min	3 (2–4)	3 (3–4)
Number of SB bouts > 60 min	2 (1–2)	1 (1–2.9)
LPA duration (min)	89.1 (81.9–99.9)	108.3 (60.9–115.7)
MVPA duration (min)	10.3 (6.8–17.3)	7.8 (2.1–59.7)

Md: medium; CI: confidence interval; *n*: number; %: percentage; SENWI: socio-ecological Nordic walking intervention; NW: Nordic walking intervention; APAFB: brief physical activity questionnaire; BMI: body mass index; HbA1c: glycated hemoglobin; SF-12: quality of life questionnaire; PCS: physical dimension; MCS: mental dimension; SB: sedentary behavior; LPA: light physical activity; MVPA: moderate–vigorous physical activity.

## Data Availability

Data will be made available upon request.

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
