# Peer review of "Assessing the Feasibility and Acceptability of a Primary Care Socio-Ecological Approach to Improve Physical Activity Adherence among People with Type 2 Diabetes: The SENWI Project"

_healthcare, 2023, doi:10.3390/healthcare11131815_

Round 1
Reviewer 1 Report
This study is focused on evaluating, through a mixed approach, the feasibility and acceptability of a socioecological intervention. Despite the adequate introduction and contextualization of the study problem that the authors carry out, the present investigation has various limitations and errors that prevent an objective evaluation of the work, some of which are mentioned below.
In summary, the authors do not use statistical parameters to refer significance.
In the same section, the authors assure that SENWI was considered acceptable and feasible without referring to any support that validates this information.
In the study design, why do the authors define their research as a pragmatic randomized controlled trial if the sampling was done for convenience? This type of sampling is the opposite of what is desired in randomized controlled trials. That is, there can be no randomization in a convenience sample because there is no equiprobability principle.
In the participants' section. Given the multifactorial nature of T2D, the description of the inclusion criteria is too broad and imprecise for the purposes of the research. The authors contemplated the time of evolution of the disease, socioeconomic status, support networks, etc.? It is important to consider these aspects to avoid biases.
In the same section, lines 127 and 128. Why did they ask the reasons for not participating in the study? This is unethical and can be understood as harassment when refusing to participate. Various laws worldwide on research with human beings explicitly state not to ask or insist on the reasons for refusing to participate in medical studies.
Lines 128. They point figure 2, before mentioning figure 1.
Lines 191, 298, 321. The authors make calls to non-existent figures.
All the citations are out of order, the authors jump from 4 to 10, from 10 to 14, from 21 to 23...
The authors use a different citation format than the journal.
There are several grammatical and spelling errors throughout the writing.
It is respectfully recommended that authors consult a person with experience in research methodology and scientific writing.
The quality of english is adequate.
Author Response
Dear reviewer,
Based on the co-authors' collective input, have carefully considered all comments and includes sever changes which we agree benefits the mansucript. To facilitate revision, all modificacions have been highlighted in the manuscript.
Below, we attach the word with all responses.
Best,
Guillem

Reviewer 2 Report
Manuscript ID: healthcare-2440995
Title: Assessing the feasibility and acceptability of a primary care socio-ecological approach to improve physical activity adherence among people with type 2 diabetes: The SENWI project.
-Why use a sample for convenience and not randomized?
- Inclusion criteria for eligible participants were T2D and without major physical limitations prescribed by the physician or any HCP. How were factors such as level of education, years of diabetes diagnosis, previous outbreaks (lifestyle, diet, physical activity), by sex controlled for?
Not having gestational diabetes, how was this criterion measured? Since in the case of women who developed gestational diabetes during their pregnancies, the risk of developing diabetes usually means that they do not associate it with a gestational diabetes event. Could the authors explain if this was considered and explain it?
In the selection criteria it is mentioned that "they could not understand the study materials and instructions", the level of adherence to understand the program or what recommendations were derived from understanding this program.
On line 123 of page 3, the abbreviation NW is mentioned for the first time, so its meaning must be fully entered, since it is done, but until page 4, on line 156.
The authors could explain with statistical argument the sample size. This doubt is because in the line 220—222 is mentioned that the sample size was based on the primary aim of examining the feasibility and acceptability of implementing a socio-ecological approach to improve PA adherence and break up long periods of SB in primary care bases in people with T2D, but the calculation is missing.
It is suggested to integrate a Cosort diagram as part of the methodology that was followed.
Measuring the effectiveness of the program was mentioned in the scope of the study, but no statistics or control variables are presented to measure it. The authors could mention what they mean by measuring the effectiveness of the study.
The focus group in what weeks or moments of the study were carried out?
In the results section, it is suggested to mention only the participants who were finally studied, that is, the 33.
Respect with the part “People with T2D perceptions” in the line 319, it follows the results about to descriptive inductive thematic analysis of the open-ended questions revealed three themes and seven subthemes between the T2D participants, but all these were not mention previously. In the line 192 to 194, it is mentioned that …to assess safety and well-being during the intervention period, patients’ perceptions about health, injury, pain, and intervention were asked every week by a specially trained professional physiotherapist in charge of delivering the NW (SNG)”. About this point, it is necessary to review this aspect are not clear of and to present them in results.
Other comments
· Changes in the sample size during the study should be described more clearly and for each group, in addition to mentioning it in the discussion.
· The main objectives of this study were: To assess the effects of the intervention, to assess a qualitative approach was used to assess the acceptability and feasibility of the intervention.
· In this aspect is necessary to give a conclusion about it with the main evidence founded and based in the statistical analysis.
· Respect to discussion needs to be improved each variable studied or outcome is necessary mentioned in this part.
· The socioecological aspects need more discussion.
· For assessing the effects of the intervention is necessary to consider statistical analysis.
· To review the references cited in the article, there are omissions, and they are not in order of appearance.
· The supplementary documents, could translate to English version?
· The tables and figures mentioned in the article are missing. The authors repeat the same documents in the supplementary and non-publishes part.
Author Response
Dear Reviewer,
Based on the co-authors' collective input, have carefully considered all comments and includes several changes which we agree benefits the manuscript. To facilitate revision, all modifications have been highlighted in the manuscript.
We add a word file with all responses and changes.
Best,
Guillem

Reviewer 3 Report
The paper titled “Assessing the feasibility and acceptability of a primary care socio-ecological approach to improve physical activity adherence among people with type 2 diabetes: The SENWI project” written by Jabardo-Camprubí and colleagues is a continuation of a paper previously published by the authors (https://doi.org/10.1186/s13063-022-06742-7). Unfortunately, this paper does not present relevant findings, and the results were not subjected to statistical analysis and were merely descriptive.
1. Authors should use superscripts to indicate the affiliation of each author with the appropriate university or research center.
2. Leave a space between the reference and the preceding text in the manuscript. Reference 10 and other references are described before reference 5 and others; the references should be described sequentially. In addition, references 11, 12, 13 , 33, 41 and 47 are not found in the manuscript. Correct all these observations.
3. Figures 2 and 3 are referred to in the text of the manuscript, but no figure was included in the manuscript or supplementary material.
Correct some grammatical errors found in the manuscript.
Author Response
Dear Reviwer,
Based on the co-authors' collective input, have carefully considered all comments and includes several changes which we agree benefits the manuscript. To facilitate revision, all modifications have been highlighted in the manuscript.
We added a word with all comment reviewer answers.
Best,
Guillem

Reviewer 4 Report
Please see document attached.

The English language can be improved in places. Some errors are identified in the reviewers comments.
Author Response
Dear reviewer,
Based on the co-authors' collective input, have carefully considered all comments and includes several changes which we agree benefits the manuscript. To facilitate revision, all modifications have been highlighted in the manuscript.
We added a word will all comment reviwer answers.
Best,
Guillem

Round 2
Reviewer 1 Report
The authors made notable changes to their manuscript.
The authors made notable changes to their manuscript
Reviewer 3 Report
Thank you for your reply.